# Transcriptomic Analysis Reveals the Involvement of Flavonoids Synthesis Genes and Transcription Factors in *Dracaena cambodiana* Response to Ultraviolet-B Radiation

Yue-E Liang [1,2,†], Hao Zhang [2,†], Jiahong Zhu [2,3], Hao Wang [2,3], Wenli Mei [2,3], Bei Jiang [1], Xupo Ding [2,3,*] and Haofu Dai [2,3,*]

1    College of Pharmacy, Dali University, Dali 671003, China
2    Key Laboratory of Research and Development of Natural Product from Li Folk Medicine of Hainan Province, Institute of Tropical Bioscience and Biotechnology, Chinese Academy of Tropical Agricultural Sciences, Haikou 571101, China
3    Hainan Institute for Tropical Agricultural Resources, Haikou 571101, China
*    Correspondence: xupoding@hotmail.com (X.D.); daihaofu@itbb.org.cn (H.D.)
†    These authors contributed equally to the work.

**Abstract:** Ultraviolet-B (UV-B) radiation is a major abiotic stress that dragon trees are exposed to during their growth and development; however, it is also an environmental signal perceived by plants that affects the flavonoid pathway. Previous studies have demonstrated that amounts of flavonoids are contained in dragon tree resin, otherwise known as dragon's blood. However, the traits and mechanism involved in the UV-B-mediated increase in flavonoids in dragon trees are still unknown. Here, we studied the response of *Dracaena cambodiana* under full solar UV-B radiation. The results showed that the contents of total flavonoids in *D. cambodiana* significantly increased after UV-B radiation exposure. Then, the transcriptome was used for determining the interactive mechanism of flavonoid accumulation and UV-B stress. Differential expression analyses identified 34 differentially expressed genes (DEGs) involved in flavonoid synthesis; specifically, 24 of the identified DEGs were significantly up-regulated after UV-B radiation exposure. In addition, 57 DEGs involved in $Ca^{2+}$/kinase sensors, 58 DEGs involved in ROS scavenging and the plant hormone pathway, and 116 DEGs transcription factors in 5 families were further identified and analyzed. Finally, we deduced the potential mechanism of UV-B-promoting flavonoid formation to neutralize ROS damage derived from UV-B radiation in *D. cambodiana* based on the gene co-expression network and previous studies from other plants. Considering that wild dragon tree populations are currently highly threatened by anthropogenic and natural stressors, the interactive studies between *D. cambodiana* plants and UV-B radiation provide valuable information toward understanding the mechanism of dragon's blood formation and help us reveal the evolution of *D. cambodiana*, with the eventual goal of aiding in the global conservation of this precious biological resource.

**Keywords:** *Dracaena cambodiana*; UV-B; RNA-seq; flavonoids metabolism; co-expression network; transcription factors; UVR8

## 1. Introduction

Dragon's blood, the rufous resin secreted from the wounded branches or stems of the *Croton* genus in the family of Euphorbiaceae, *Dracaena* genus in the family of Asparagaceae, *Daemonorops* genus in the family of Palmaceae, and *Pterocarpus* genus in the family of Fabaceae [1–3], is a precious traditional medicine which has been widely used in many countries. The evidential origins of dragon's blood were first referenced by the Periplus Maris Erythraei in the mid-first century BC [4]. In China, the first recorded use of dragon's blood was listed in the "Lei Gong Pao Zhi Lun" during the Liu-Song period of the Northern

and Southern dynasties and it has been used for more than 1500 years. Chinese traditional pharmacists have broadly utilized dragon's blood since ancient times as a traditional Chinese medicine to promote blood circulation and relieve inflammation and pain and astringent sores. Modern pharmacological studies have demonstrated that dragon's blood also has anti-inflammatory, anti-bacterial [5,6], antitumor [7], and hypoglycemic [8] effects, as well as cardiovascular protection [9] and anti-oxidation properties [10,11]. Satisfactory medicinal effects resulting from wild *Dracaena* resources have been excessively overexploited in the past three decades [12]. In addition, dragon's blood formation in the wild requires external stimulation and, therefore, takes 10 years or even decades. Previous studies have found that dragon's blood exhaustion can be produced by alien induction methods such as fungi inoculation and chemical induction [13,14]. However, the detailed molecular mechanism of dragon's blood formation still remains mysterious.

Flavonoids and steroidal saponins are vital biological components and the main contributors to the medicative function in dragon's blood [15–18]. More specifically, flavonoids present antioxidant, anti-inflammatory, antibacterial, and antidiabetic activities, and have favorable biochemical effects on cardiovascular disease and atherosclerosis [19,20]. Flavonoids, the largest group of plant phenols, contain a basic skeleton of 2-phenylchromone that can be classified into six subgroups according to a diversity of substituents, including chalcones, flavonols, flavandiols, flavones, anthocyanins, and proanthocyanidins or condensed tannins [21]. Flavonoids are ubiquitous in the plant kingdom and are usually detected abundantly in flowers, leaves, and seeds [21]. Plant flavonoids play important roles in plant growth and development, and in the responses of plants interacting with biotic and abiotic stresses [22]. The structural types and amount or content of flavonoids in different plants are various, resulting in diverse and complex flavonoid biosynthesis, transportation, and regulation, especially for structurally specific flavonoids and their derivative polymers [23,24]. In general, plant flavonoids are generated through the phenylpropanoid pathway and the enzymes involved; in the first three steps, phenylalanine ammonia lyase (PAL), cinnamate 4-hydroxylase (C4H), and 4-coumaroyl-CoA lyase (4CL) gradually catalyze phenylalanine into 4-coumaroyl-CoA [25]. Then, chalcone synthase (CHS) serves as the first enzyme and the rate-limiting step for various phases of flavonoid production, and produces an articulated skeleton for subsequent flavonoid synthesis [26]. Chalcone isomerase (CHI) subsequently generates flavanones which serve as essential intermediate metabolites and the key branch points in the biosynthesis pathway of flavonoids [27,28]. Flavanone 3-hydroxylase (F3H) is the pivotal enzyme in dihydroflavonol biosynthesis, which can catalyze the dydroxylation of flavanones at position C-3 and produce the common precursors for flavonol, anthocyanin and proanthocyanin [29,30]. Dihydroflavonol 4-reductase (DFR) can catalyze substrates of similar structure to formulate a hydroxyl group at position C-4 of ring C and has a crucial role in anthocyanidin and proanthocyanidin synthesis [31]. Many different enzymes involved in producing different flavonoid subclasses or their associated derivatives increase their diversity and contribute to multiple biological activities [32]. In addition, most flavonoids generally act as plant sunscreen and antioxidants to prevent or limit damage from UV stress [33]. Flavonoid biosynthesis is activated and accumulated in epidermis cells to induce protective responses when plants are exposed to UV stress [26].

Ultraviolet-B (UV-B) radiation exposure is a serious challenge for the survival of *Dracaena* species and their population dispersion regarding their growth conditions. UV-B radiation exposure increases levels of reactive oxygen species (ROS), and has negative effects on plant cells [34]. As an abiotic environmental stress, reactive oxygen species (ROS) caused by UV-B radiation in organisms can damage cellular constituents and macromolecules, impairing photosynthesis and respiratory action, thereby influencing plant growth, development, and morphology [33]. Various sophisticated systems have evolved and been developed by plants as protection against UV-B radiation stress. UV-induced morphological changes, including alterations in leaves, stems, roots, and inflorescence, to shading, reflecting, and decreased penetration of UV-B radiation constitute a common strategy [35,36]. However,

one of the most effective defense strategies for plants is the enhancement of the antioxidant system and accumulation of various UV-B-absorbing compounds to quench reactive oxygen and nitrogen species, especially flavonoids [37]. Flavonoids can absorb light at 280–340 nm and act as antioxidants, scavenging ROS to help plants survive UV-B radiation damage [38]. The overaccumulation of kaempferol and quercetin (both flavonoids) is crucial for enhanced UV radiation tolerance in rice [39]. Due to the lack of flavonoid content and oxidative stress, pea plants were more impressionable to UV-B radiation in high air humidity than in moderate air humidity [40]. Anthocyanin and flavonols play essential roles in apple plants by combating the harmful effects of UV-B radiation stress, and more flavonols are produced by apple fruit peels exposed to higher levels of UV-B radiation [41,42]. The accumulation of "sunscreen" flavonoids to prevent or limit damage under UV-B radiation stress is widely common among plants. Although those flavonoids interacting with UV-B radiation have been studied in various plants—even in specific plant species [43]—reports about flavonoid synthesis in *Dracaena* species under UV-B radiation have attracted little attention.

As the specific metabolites are complex, dragon's blood from *Dracaena* plants presents a structural diversity of flavonoids, and their contents are higher than those in other species, which might be due to the effects of the ecological environment. *Dracaena* species are used for dragon's blood production, and usually grow in subtropical and tropical regions [15,44]. *Dracaena cambodiana* is the main plant resource for dragon's blood production in Southeast Asia and China and can usually be found in barren land, for instance, on cliffs, in stone cracks, or on desert islands [17]. Under these special ecological environments, *D. cambodiana* experiences periodic exposure to high salinity, high temperature, and high ultraviolet-B radiation. Despite these discoveries, the survival mechanism of *D. cambodiana* has not been well elucidated at the transcriptome level, especially in response to long periods and high doses of UV-B radiation.

In this study, we performed a transcriptomic analysis of *D. cambodiana* plants under UV-B radiation based on the phenomenon of UV-B radiation promoting total flavonoid content in the stems of *D. cambodiana*, aiming to determine the DEGs involved in the UV-B radiation stress response, as well as flavonoid biosynthesis and regulation in *D. cambodiana* exposed to UV-B radiation, thereby gaining some insight into the relationship between *Dracaena* species and UV-B radiation. These results provide a theoretical foundation for the further study of flavonoid biosynthesis pathways in *D. cambodiana*, and the roles of transcription factors in dragon's blood formation and dragon trees' responses to UV-B stress, enhancing our understanding of the survival and longevity mechanism of *Dracaena* species.

## 2. Materials and Methods

### 2.1. Plant Materials

Three-year-old *D. cambodiana* plants were cultured in the greenhouse of the Institute of Tropical Biosciences and Biotechnology, Chinese Academy of Tropical Agricultural Sciences (19°59′ N, 110°19′ E). Stem samples were collected after 0 h, 3 h, 12 h, and 24 h of UV irradiation. The samples were frozen in liquid nitrogen immediately after being weighed. The frozen samples were stored at −80 °C in a refrigerator until total flavonoid and RNA extraction.

### 2.2. The Activities of CAT and POD, the Contents of $H_2O_2$, and Total Flavonoid Determination

The activities of antioxidant enzymes catalase (CAT) and peroxidase (POD) were measured using the ultraviolet absorption method. Fresh 200 mg samples of *D. cambodiana* stems after UV-B radiation treatment were harvested at 0, 3, 12, and 24 h, and were homogenized immediately in 2.0 mL of 60 mM phosphate buffer (pH 7.8) within an ice bath. A Catalase (CAT) Activity Assay Kit (Boxbio, Beijing, China) and a Peroxidase (POD) Activity Assay Kit (Boxbio, Beijing, China) were used to determine the activities of CAT and POD of *D. cambodiana* stem samples using the ultraviolet absorption method according to the manufacturer's instructions. The absorbances for CAT and POD kits were monitored at 240 nm (extinction coefficient: 0.036 mM$^{-1}$·cm$^{-1}$) and 470 nm (extinction coefficient:

$26.6 \text{ mM}^{-1} \cdot \text{cm}^{-1}$), respectively, using an enzyme-labeling measuring instrument (ELX-800, Bio-Tek, USA). $H_2O_2$ content was assayed by monitoring the absorbances of the titanium–peroxide complex at 415 nm using the $H_2O_2$ Content Assay Kit ($H_2O_2$-1-Y, Comin, Suzhou, China). Total flavonoid contents of the abovementioned samples were determined using a method described in previous studies [45]. Aluminum trichloride ($AlCl_3$), specifically, reacts with flavonoids, creating a flavonoid–Al complex that presents significant absorption at 420 nm. The plant samples were ground in liquid nitrogen, and then samples used to determine total flavonoid contents were gained via extraction and incubation conducted 3 times within one hour, using ethyl acetate. During each incubation period, the extractives in the tubes were mediated 5–6 times. At the end of the incubation process, the extractives were filtered using qualitative filter paper and then rotatory evaporated at 37 °C to obtain total flavonoid extractive. Finally, the flavonoid extract was dissolved into 1 mL methanol. An appropriate amount of methanol extract was taken and mixed with 10% aluminum trichloride to reach a final concentration of 1.00%. Absorbance was detected at 420 nm and the fresh weights of *D. cambodiana* stems were used to normalize the absorbance values. Each treatment was performed by using five independent biological replicates and three technical replicates. The results were analyzed using an analysis of variance (ANOVA) test.

### 2.3. RNA Extraction

The RNAprep Pure Plant plus Kit (Tiangen, Beijing, China) was used for total RNA extraction. The quality and concentration of total RNA were monitored by 1% agarose gel electrophoresis, as well as the system of Bioanalyzer 2100 (Agilent, CA, USA), Nanodrop One™ (Thermo Scientific, Wilmington, DE, USA) and Qubit 2.0 (Invitrogene, Carlsbad, CA, USA).

### 2.4. cDNA Library Construction and Illumina Sequencing

The NEBNext Ultra RNA Library Prep Kit (NEB, Ipswich, MA, USA) was utilized to create sequencing libraries according to the instructions of the manufacturer. Initially, mRNA was isolated from total RNA, utilizing poly-T oligo-linked magnetic beads. Subsequently, fragmentation of the mRNA was conducted using divalent cations in 5× Next First-Strand Synthesis Reaction Buffer (NEB, USA). First-strand cDNA was generated by using M-MuLV Reverse Transcriptase and random hexamer primers, whereas, for DNA polymerase I, dNTPs and RNase H were employed for the generation of second-strand cDNA. After adenylation and adaptor ligation of the DNA fragments at 3′ ends, the library fragments were purified with the AMPure XP system (Beckman Coulter, Boston, MA, USA) to select cDNA fragments 370–420 bp in length. The cDNA libraries were constructed by PCR enrichment and were then assessed on an Agilent Bioanalyzer 2100 system. Subsequently, the cDNA library was sequenced on the Illumina HiSeq X-10 platform. RNA-Seq included two replicates for each treatment.

Clean reads were obtained by removing adaptors and low-quality sequences. Then, non-redundant transcripts were assembled with clean reads using Trinity v 2.2.0 with default parameters, a specific software for the de novo assembly and transcriptome analysis of the species without the use of whole genome sequencing and using short illumine reads.

### 2.5. Unigenes Annotation and Differentially Expressed Gene Analysis

The methods of unigene annotation and enrichment were sourced from our previous studies [46]. Unigene sequences were firstly annotated with Pfam, NR COG, KOG, Swiss-Prot, and eggNOG databases [46,47]. Subsequently, the annotated results were enriched with the Gene Ontology (GO) database by Blast2GO and were classified into three subfamilies, namely cellular component (CC), biological process (BP), and molecular function (MF). Pathway enrichment was conducted using the KEGG Automatic Annotation Server (KAAS) from the Kyoto Encyclopedia of Genes and Genome (KEGG) database. Gene expression levels of unigenes were calculated and are presented as FPKM values based on the count

values from RSEM assessments after mapping clean reads on the final unigene assembly using Bowtie [13]. False discovery rate (FDR) < 0.001 and an estimated absolute log2 fold change (log2 FC) $\geq$ 1 were the thresholds for the selection of differentially expressed genes (DEGs).

*2.6. Correlation Networks*

The FPKM values of key DEGs involved in flavonoid biosynthesis (*CHS*, *F3H*, and *DFR*) and TFs (*WRKY*, *bZIP*, *MYB*, *bHLH*, and *WD40*) were extracted, and Pearson's correlation coefficient (*r*) was employed to examine their co-expression. The gene–TF pairs (*p*-value < 0.05 and $r^2 \geq 0.8$) were selected as significant co-expression patterns and we present the correlation expression networks using Cytoscape (v 3.8.0) [48].

*2.7. Quantitative Analysis of Gene Expression*

The cDNA, reverse-transcribed from total RNA by using the first-strand cDNA synthesis Kit (Tiangen, Beijing, China) and quantified to 1 μg/μL, was diluted 20 times and used as the cDNA template for real-time PCR. Nine DEGs of representation, namely *CHS*, *CHI*, *DFR*, *F3H*, *UVR8*, *MYB*, *bHLH*, *WD40*, and *P450*, were selected for verification with RT-qPCR and specific primers were designed with Primer 3.0, whereas *β*-actin was used as an endogenous control gene (Table S1). RT-qPCR was performed with a total volume of 20 μL: 1 μL of cDNA temple, 10 μL of SYBR Green Real-time PCR Master Mix, 8.2 μL of RNA-free water, and 0.4 μL of each primer. The reaction conditions on the CFX96 Real-Time system (Bio-Raid CFX) were 95 °C for 5 min, and 42 cycles of 95 °C for 5 s and 60 °C for 30 s. Reproducibility and reliability were ensured with three technical and biological replicates for each plant. To measure the expression degrees of genes as fold changes between the reference sample and the selected sample, the $2^{-\Delta\Delta CT}$ method was employed for calculating relative expression values [46].

## 3. Results

*3.1. Enzyme Activities of CAT and POD, and Contents of $H_2O_2$ and Total Flavonoids in the Stems of D. cambodiana after UV-B Radiation Treatment*

The dynamic profiles of $H_2O_2$ content, CAT activity, POD activity, and total flavonoid content in the stems of *D. cambodiana* after 0, 3, 12, and 24 h of UV-B radiation treatment were determined. The $H_2O_2$ contents of *D. cambodiana* were significantly increased after UV-B radiation treatment, which suggested that UV-B radiation promoted $H_2O_2$ accumulation (Figure 1a). However, CAT activities were not significantly different (Figure 1b), suggesting that CAT was not the key enzyme in neutralizing $H_2O_2$ damage in this process. Moreover, the UV-B radiation treatment demonstrated that the activities of POD significantly decreased in the stems of *D. cambodiana*, and decreased gradually, suggesting that they were feedback-inhibited by their catalysates (Figure 1c). Considering the catalysis function of POD in plants, these catalysates were possibly flavonoids. Subsequently, total flavonoid content in the stems of *D. cambodiana* significantly increased after UV-B irradiation when compared with the sample without UV-B irradiation (Figure 1d), especially when considering the total flavonoid content in the 24-h sample, which increased nearly threefold when compared with the control check. However, there was no significant change between the 3-h and 12-h samples. This result suggested that it can be tentatively shown that UV-B irradiation stimulates flavonoid accumulation in the stems of *D. cambodiana*.

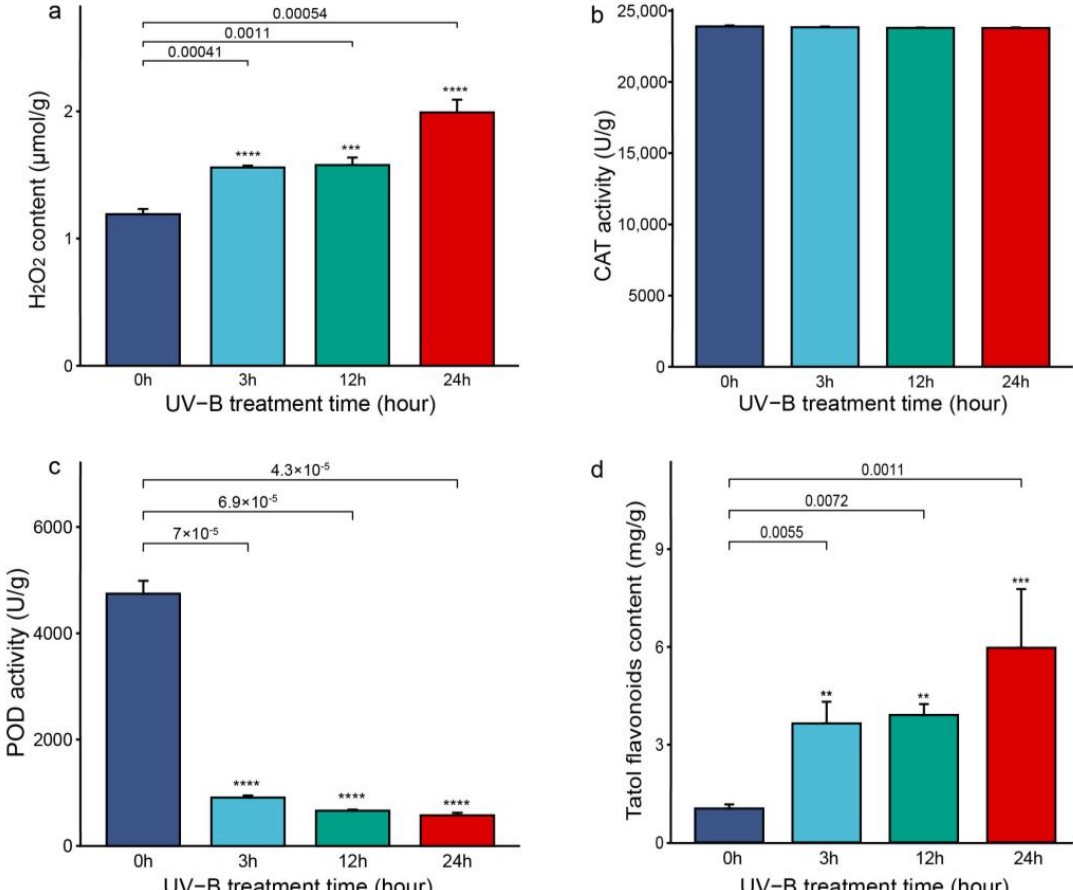

**Figure 1.** Enzyme activities and contents of $H_2O_2$ and total flavonoids in the stems of *Dracena. cambodiana* after different UV-B radiation treatment times. (**a**) $H_2O_2$ content, (**b**) CAT activity, (**c**) POD activity, (**d**) Total flavonoid content. Asterisks indicate statistically significant differences when compared with the 0-h sample, ** $p < 0.01$, *** $p < 0.001$, **** $p < 0.0001$.

### 3.2. Assembly and annotation of D. cambodiana Transcriptome

By sequencing on the Illumina Hiseq X-10 platform, clean reads amounting to 58.63 Gb were produced after removing low-quality reads (Table S2). Additionally, Q20 and Q30 averaged 94% and 95%, respectively, whereas the GC contents averaged 46%. Finally, 103,643 unigenes and 277,117 transcripts were assembled. The total nucleotide numbers were 86,463,273 bp and 25,848,265 bp, and associated N50 counts were 1433 and 1718, respectively (Table S3). Transcripts and unigenes of length 200–500 bp were the most abundant (Figure 2a).

In total, 103,643 unigenes were functionally annotated in eight public databases (Nr, SwissProt, eggNOG, COG, KOG, Pfam, GO, and KEGG) (Table 1). In the annotation results, 40,641 unigenes were annotated and accounted for 39.21% of the total number of unigenes. Among them, 9811 unigenes had COG homologs, 15,548 had GO homologs, 15,601 had KEGG homologs, 21,153 had KOG homologs, 22,671 had Pfam homologs, 24,608 had Swissprot homologs, 34,553 had eggnog homologs, and 38,869 had Nr homologs. Among these annotated unigenes, 16,036 unigenes had lengths between 300 and 1000 bp and the lengths of the other 17,406 unigenes were greater than 1000 bp. A total of 38,869 unigenes were matched within the Nr database, which accounted for 95.64% of the total of annotated unigenes. The species most represented in Nr homologous analysis was *Asparagus officinalis* (Figure 2b), accounting for 46.33% of all annotated unigenes. The other homologous species in the Nr database were *Elaeis guineensis* (2103, 5.41%), *Phoenix dactylifera* (1852, 4.76%), *Musa acuminata* (924, 2.38%), *Zea mays* (819, 2.11%), *Oryza sativa* (808, 2.08%), *Ananas comosus* (782, 2.01%), *Vitis vinifera* (552, 1.42%), *Dendrobium catenatum* (486, 1.25%), and *Cajanus cajan*

(337, 0.87%). A percentage (31.38%) of unigenes were annotated to other species (Figure 2b). *A. officinalis* and *D. cambodina* both belong to the family of Asparagaceae, which suggests that the transcriptome assembly and annotation were accurate.

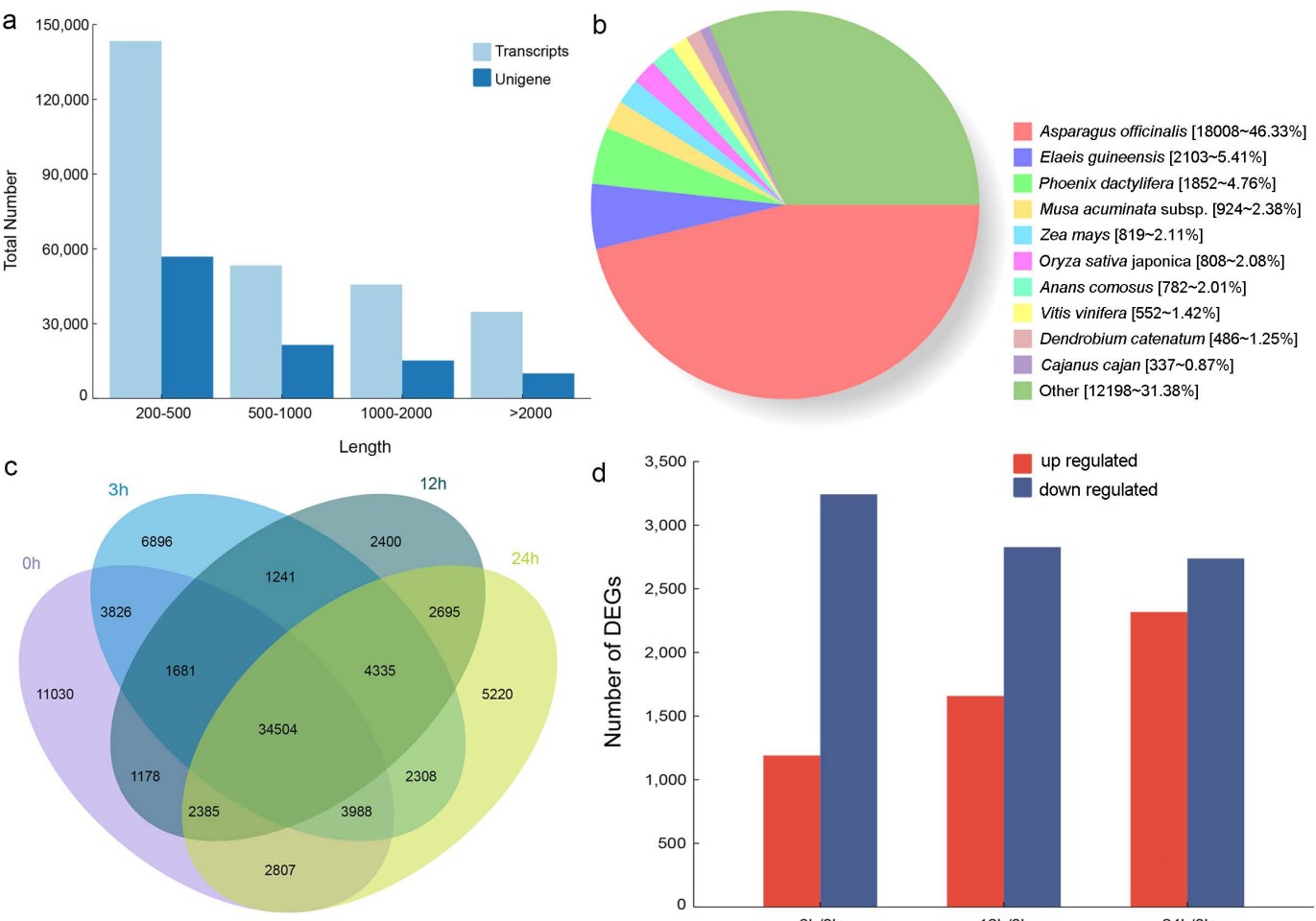

**Figure 2.** Statistics of unigenes and differentially expressed genes. (**a**) Length distribution of unigenes and transcripts; (**b**) Top 10 species distribution based on Nr database of final unigenes; (**c**) The number of unique unigene expressions among four treatments; (**d**) The number of up-regulated and down-regulated DEGs in the groups of 3 h vs. 0 h, 12 h vs. 0 h, and 24 h vs. 0 h. Significance of DEGs was identified with the parameters of FDR (false discovery rate) $\leq 0.01$ and FC (fold change) $\geq 2$.

**Table 1.** The annotation of unigenes of *Dracaena cambodiana* in public databases.

| Database | Number of Annotated Unigenes | $300 \leq$ Length $< 1000$ | Length $\geq 1000$ |
|---|---|---|---|
| COG | 9811 | 2558 | 5987 |
| GO | 15,548 | 5759 | 6859 |
| KEGG | 15,601 | 5896 | 7222 |
| KOG | 21,153 | 7493 | 10,341 |
| Pfam | 22,671 | 6179 | 14,123 |
| Swissprot | 24,608 | 9003 | 12,284 |
| eggNOG | 34,553 | 12,779 | 16,041 |
| Nr | 38,869 | 15,201 | 17,079 |
| Total | 40,641 | 16,036 | 17,406 |

### 3.3. Differentially Expressed Genes (DEGs) in the Stems of D. cambodiana after UV-B Irradiation

To determine unigene expression levels in each treatment, the RNA-Seq clean reads from each library were aligned to the final assembly. A total of 34,504 genes were expressed

in 0 h, 3 h, 12 h, and 24 h samples, including 11,030, 6896, 2400, and 5220 unigenes, respectively (Figure 2c). The differentially expressed genes in 3 h and 0 h (3 h vs. 0 h), 12 h and 0 h (12 h vs. 0 h), and 24 h and 0 h (24 h vs. 0 h) groups were screened according to the fold change (FC) $\geq$ 2 and false discovery rate (FDR) $\leq$ 0.01. A total of 1254 DEGs were detected in the 3 h vs. 0 h group, of which 1190 were up-regulated and 3242 were down-regulated. In the 12 h vs. 0 h group, 4485 unigenes were found, including 1658 up-regulated and 2827 down-regulated DEGs. A total of 5085 differentially expressed genes were found in the 24 h vs. 0 h group, including 2317 up-regulated and 2738 down-regulated DEGs (Figure 2d).

### 3.4. Functional Enrichment Analysis of DEGs

The differential genes were enriched using Gene Ontology (GO) terms and Kyoto Encyclopedia of Genes and Genomes (KEGG) pathways for functional classification (Table S4). In the 3 h vs. 0 h treatment group, 1250 DEGs were enriched by GO terms. Among them, molecular function was the most representative, with 1035 DEGs. The remaining biologi-cal processes were enriched to 928 unigenes and 683 unigenes for cell components. (Figure 3a). Additionally, 796 DEGs were annotated in the KEGG pathway database (Figure 3b). Among them, the top five GO terms enriched with statistical significance in this group were metabolic process (GO: 0008152, 737), catalytic activity (GO: 0003824, 655), binding (GO: 0005488, 617), cellular process (GO: 0009987, 609), and cell (GO: 0005623, 561), and the top five in the KEGG database were carbon metabolism (ko01200, 94), biosynthesis of amino acids (ko01230, 76), photosynthesis (ko00195, 59), oxidative phosphorylation (ko00190, 55), and photosynthesis–antenna proteins (ko00196, 49) (Table S5). Further, 1591 DEGs were enriched in the 12 h vs. 0 h group, of which 1251 were within molecular functions, 1179 belonged to biological processes, and 898 were cellular components (Figure 3c), and 953 DEGs were enriched into KEGG (Figure 3d). Among them, the first five enriched GO terms and KEGG pathways were metabolic process (GO: 0008152, 942), catalytic activity (GO: 0003824, 821), cellular process (GO: 0009987, 754), cell (GO: 0005623, 720), single-organism process (GO: 0044699, 736), carbon metabolism (ko01200, 123), biosynthesis of amino acids (ko01230, 107), photosynthesis (ko00195, 81), phenylpropanoid biosynthesis (ko00940, 62), and photosynthesis–antenna proteins (ko00196, 60), respectively (Figure 3d). In addition, 1767 differential unigenes were enriched to the 24 h vs. 0 h group, of which 1404 unigenes were enriched to molecular functions, 1325 to biological processes, and 987 to cellular processes (Figure 3e). A total of 1060 DEGs in this group were annotated into the KEGG pathway database (Figure 3f). In this group, the top five GO terms and KEGG pathways were metabolic process (GO: 0008152, 1092), catalytic activity (GO: 0003824, 953), cellular process (GO: 0009987, 871), single-organism process (GO: 0044699, 824), cell (GO: 0005623, 796), carbon metabolism (ko01200, 138), biosynthesis of amino acids (ko01230, 110), photosynthesis (ko00195, 82), phenylpropanoid biosynthesis (ko00940, 71), and carbon fixation in photosynthetic organisms (ko00710, 71), respectively. In KEGG enrichment, most of the DEGs were concentrated in metabolic pathways. Further, some DEGs were also enriched in pathways of genetic information processing, environmental information processing, cellular processing, and organismal systems, for instance, plant hormone signal transduction (ko04075), ubiquinone and other terpenoid-quinone biosynthesis (ko00130), plant–pathogen interaction (ko04626), and protein processing in endoplasmic reticulum (ko04141).

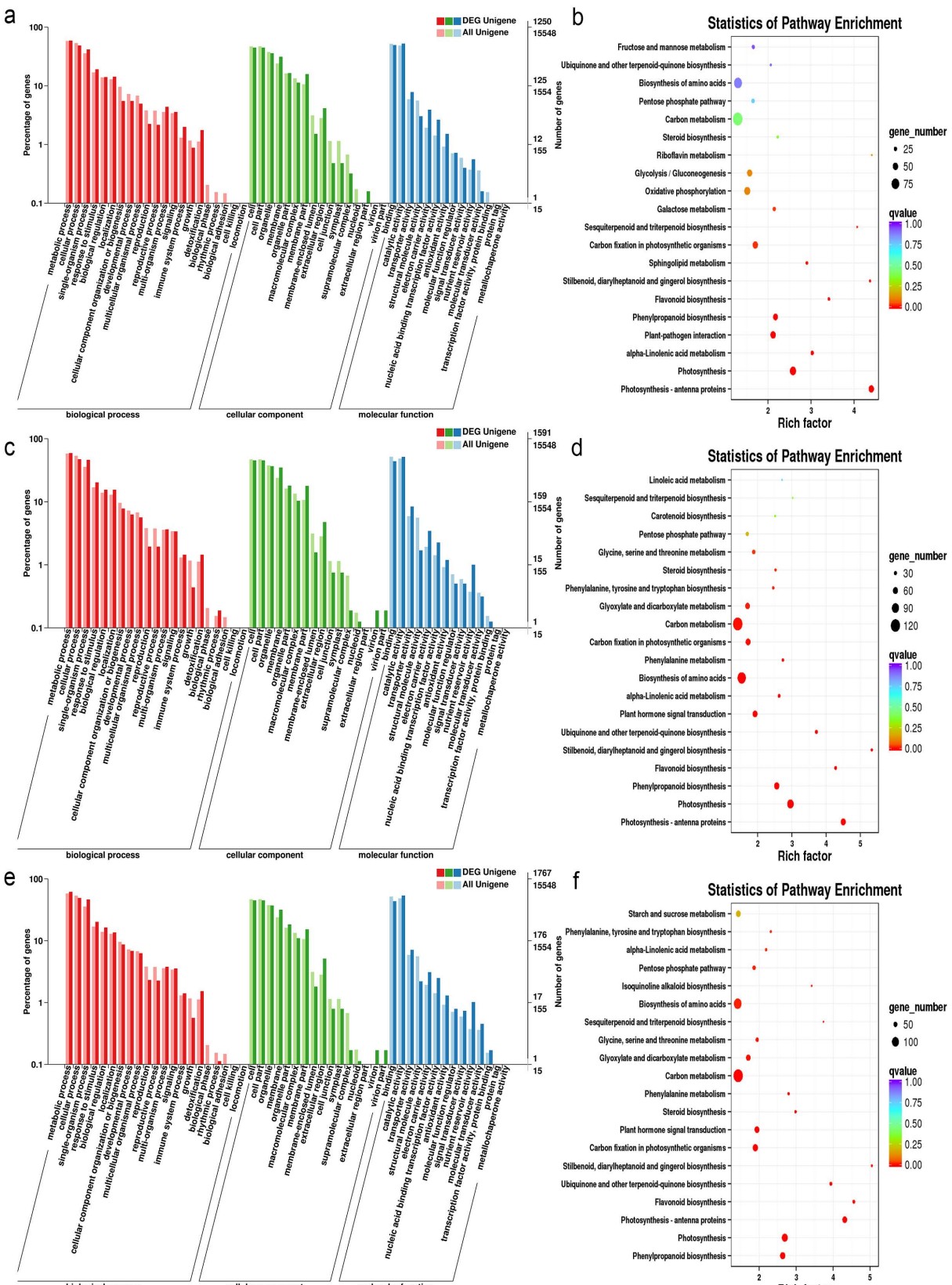

**Figure 3.** The enrichment of DEG annotation using GO terms and KEGG pathway. (**a**,**b**) GO and KEGG pathway classifications of DEGs in 3 h vs. 0 h group; (**c**,**d**) GO and KEGG pathway classifications of DEGs in 12 h vs. 0 h group; (**e**,**f**) GO and KEGG pathway classifications of DEGs in 24 h vs. 0 h group.

### 3.5. DEGs Involved in UV-B Response and Signal Cascade Transduction

According to GO annotation analysis in gene function annotation, 55 unigenes classified into six GO terms were associated with UV function (Table S8), including UV protection (GO: 0009650, 3), anthocyanin accumulation in tissues in response to UV light (GO: 0043484, 17), cellular response to UV-B (GO: 0071493, 1), response to UV (GO: 0009411, 9), response to UV-B (GO: 0010224, 22), and response to UV-C (GO: 0010225, 4). A total of 22 unigenes presented potential functions of response to UV-B. The unigenes involved in the GO terms of UV protection and response to UV-C presented little or no expression profiles. Subsequently, the number of unigene responses to UV or UV light was 26, of which 7 unigenes presented as DEGs. In brief, 18 unigenes within 55 unigenes were identified as DEGs. Nine unigenes were classified into responses to UV, and two DEGs contained the motif of UVR8 function (Figure 4a). In the *D. cambodiana* transcriptome, 2 NO-related DEGs (2 nitrate reductase, NR), 24 ROS-related DEGs (1 NADPH oxidases, NOX; 3 polyphenol oxidase, PPO; 7 peroxidase, POD; 7 ascorbate peroxidase, APX; 4 catalase, CAT and 2 superoxide dismutase, SOD), 12 JA-related DEGs (6 lipoxygenase, LOX; 1 alternative oxidase, AOX and 5 allene oxide cyclase, AOC), 20 ethylene-related DEGs (5 1-aminocyclopropane-1-carboxylate synthase, ACS and 15 1-aminocyclopropane-1-carboxylate oxidase, ACO) (Figure 4b), 20 MAPK-related DEGs (6 mitogen-activated protein kinase, MAPK; 6 mitogen-activated protein kinase kinase, MAPKK and 8 mitogen-activated protein kinase kinase kinase, MAPKKK), and 37 Ca$^+$ signaling transduction-related DEGs (16 calmodulin, CaM and 21 calcium-dependent protein kinase, CDPK) were screened (Figure 4c and Table S7), which may lead to the accumulation of flavonoids in *D. cambodiana* through UV-B induction.

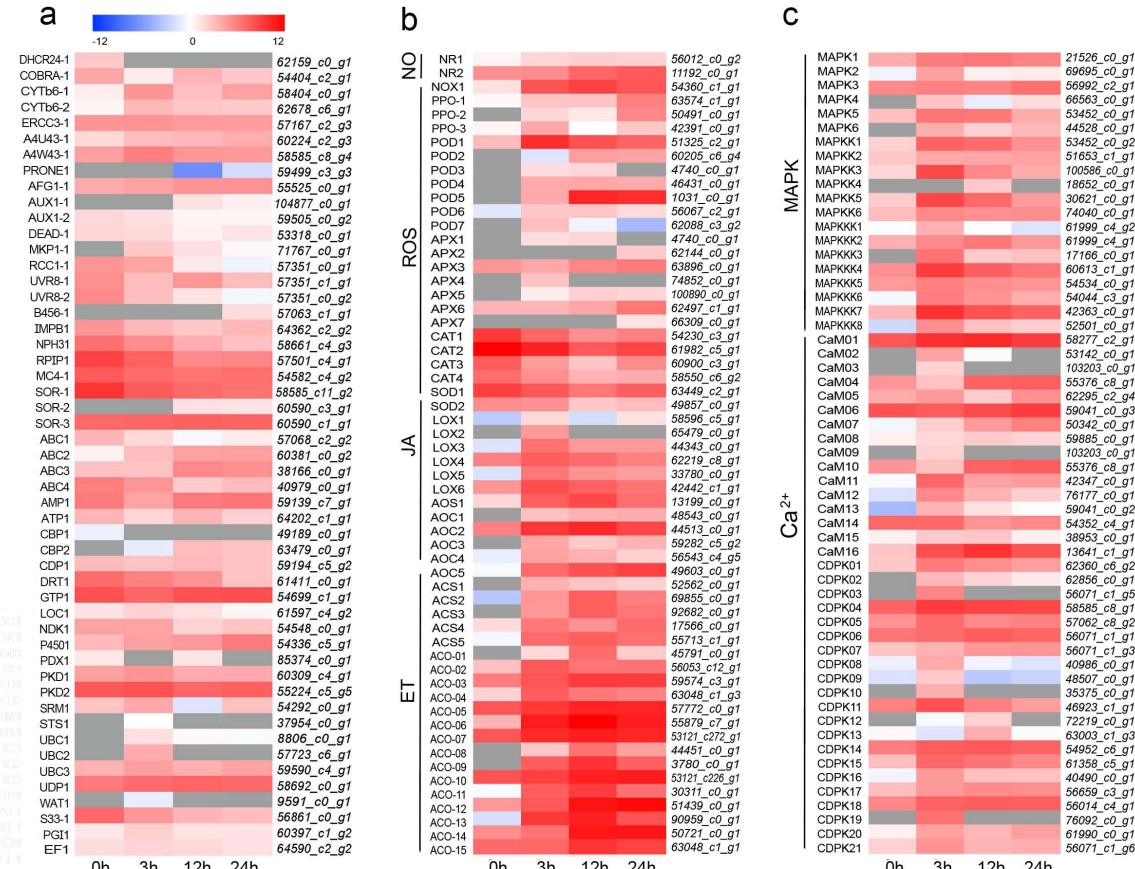

**Figure 4.** Expression profiling of DEGs in UV-B irradiation response and signal cascade transduction. (**a**) Expression pattern of DEGs involved in UV irradiation response; (**b**) Expression profiling of DEGs involved in signal cascade transduction; (**c**) Expression profiling of DEGs involved in MAPK and Ca$^{2+}$ signal cascade transduction.

### 3.6. DEGs Involved in Flavonoid Biosynthesis

Flavonoids and their derivatives are the major components of *D. cambodiana* dragon's blood. The biosynthetic pathway of flavonoids in many plants has been studied and reported. In this section, we aimed to obtain the expression profiles of crucial genes involved in flavonoid synthesis. A brief schematic of flavonoid biosynthesis was constructed (Figure 5a). Unigenes related to flavonoid synthesis were detected from the transcriptome of *D. cambodiana* treated with UV-B irradiation and 34 unigenes were identified as DEGs, including three *PAL*, one *C4H*, ten *4CL*, four *CHS*, one *CHI*, one *F3H*, one *FLS*, one *FDR*, and two *LAR*, which were substantially up-regulated after UV treatment, consistent with flavonoid accumulation (Figure 5b). In addition, cytochrome P450 may produce a range of different secondary metabolites through hydroxylation and monooxygenation reactions and may participate in the oxidation reaction of plant flavonoid biosynthesis, which can promote the flavonoid compounds to have a variety of chemical structures [49]. In our study, a total of 155 *P450*-related unigenes were identified in the Nr database, of which 85 unigenes were identified as DEGs, including 37 up-regulated DEGs and 48 down-regulated DEGs, which may be related to the modification reaction of flavonoid synthesis in *D. cambodiana* (Table S10). The co-expression of these structural genes and modified enzymes in the synthesis of the phenylpropane metabolic pathway might promote the accumulation of flavonoids and the formation of dragon's blood in *D. cambodiana*.

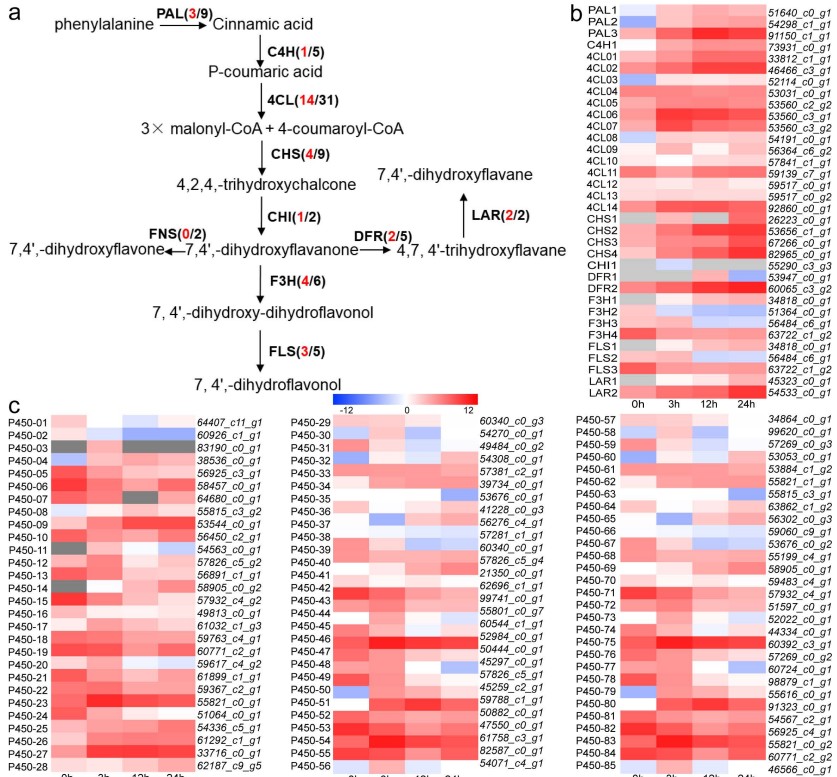

**Figure 5.** Expression profiling of DEGs in flavonoid biosynthesis. (**a**) Proposed pathway for flavonoid biosynthesis in *D. cambodiana*. The black and red numbers in brackets following each gene name indicate the total number of unigenes and DEGs annotated to that gene, respectively. Enzyme abbreviations: PAL, phenylalanine ammonia lyase; C4H, cinnamate 4-hydroxylase; 4CL, 4-coumarate CoA ligase; CHS, chalcone synthase; CHI, chalcone isomerase; F3H, flavanone 3-hydroxylase; FLS, flavonol synthase; DFR, dihydroflavonol 4-reductase; LAR, leucoanthocyanidin reductase; FNS, flavone synthase; (**b**) Expression levels of the DEGs coding key enzymes involved in flavonoid biosynthesis pathways; (**c**) Differential expression unigene of Cytochrome P450 (CYP). Blue and red colors are used to represent low-to-high expression levels, and color scales correspond to the mean-centered log2-transformed FPKM values.

### 3.7. Candidate Transcription Factors Involved in Flavonoid Biosynthesis

Transcription factors (TFs) play crucial roles in regulating the biosynthesis and transport of various metabolites [50], including flavonoid biosynthesis [51]. Based on our transcriptomic results, 1508 unigenes were identified as TFs and 411, 855 unigenes were classified as transcription regulators (TRs) or protein kinase (PKs), respectively (Table S11). Previous studies have demonstrated that at least five different transcription factors (MYB, bHLH, WD40, WRKY, and bZIP) regulate flavonoid biosynthesis in plants. Transcription factors such as R2R3-MYB, bHLH, and WD40 can work individually or orchestrate with others (a ternary complex of MYB–bHLH–WD40) to regulate various enzymatic steps involved in flavonoid biosynthetic pathways. Regarding these five TF families,116 TFs were detected as DEGs in the stems of *D. cambodiana* after UV treatment, including 9 *bZIP* (5 up-regulated and 4 down-regulated), 26 *MYB* (12 up-regulated and 14 down-regulated), 30 *bHLH* (11 up-regulated and 19 down-regulated), 63 *WRKY* (39 up-regulated and 24 down-regulated), and 12 *WD40* (2 up-regulated and 10 down-regulated) TFs in the stems of *D. cambodiana* after UV treatment (Figure 6). These differentially expressed TFs may be involved in regulating flavonoid biosynthesis in *D. cambodiana*.

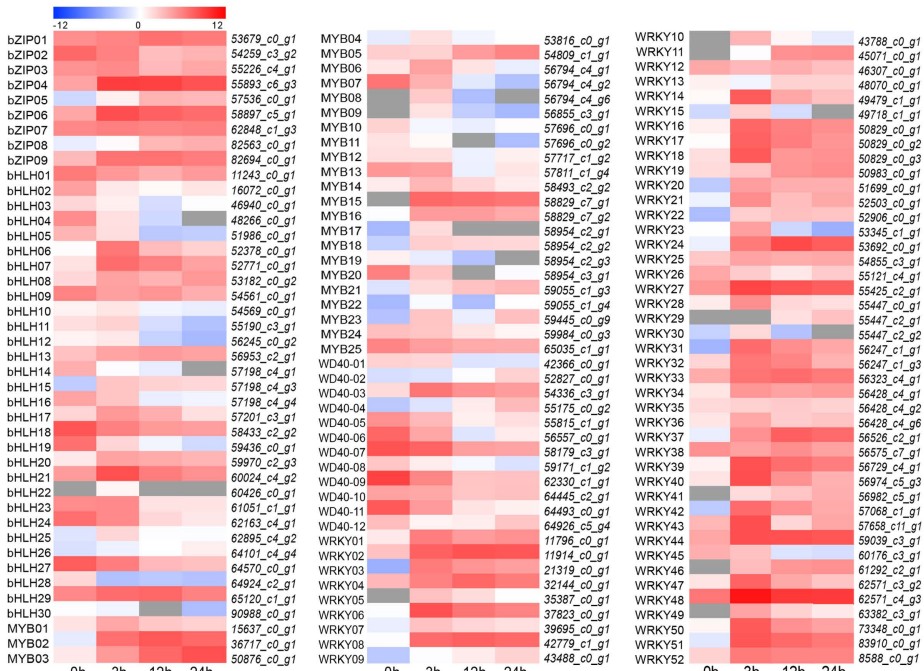

**Figure 6.** Expression profiling of DEGs from the TF gene families of *MYB*, *bHLH*, *WD40*, *bZIP*, and *WRKY*.

### 3.8. The Interactive Network of UV-B Regulates Flavonoid Accumulation in D. cambodiana

The co-expression network was constructed with 9 important flavonoid biosynthesis genes, 72 TFs (16 MYB, 10 WD40, 21cbHLH, 19 WRKY, and 6 bZIP TFs), and 2 UVR8 to find the potential regulators involved in flavonoid biosynthesis and UV-B response. A total of 46 TFs were only strongly correlated with *UVR8-1*, *UVR8-2*, *F3H2*, and *F3H4*, including 6 TFs that were significantly correlated with *UVR8-1*. Interestingly, *UVR8-2* was also positively correlated with *F3H2* and *F3H4*. In addition, another 24 TFs were correlated with 7 DEGs involved in flavonoid biosynthesis (*CHS1-4*, *DFR2*, *F3H1*, and *F3H3*), and the last two TFs (*WD40-7* and *WRKY45*) showed correlation with 7 DEGs (*CHS1*, *CHS2*, *CHS3*, *CHE4*, *DFR2*, *F3H2*, and *F3H3*) and *UVR8-2*. In this network, *UVR8-1* showed a positive relationship with 3 TFs (*bHLH09*, *WRKY12*, and *WRKY38*) and a negative relationship with 5 TFs (*WRKY34*, *bHLH08*, *bZIP7*, *MYB22*, and *MYB23*), whereas *UVR8-2* showed a positive relationship with 28 TFs and a negative relationship with 11 TFs. *CHS1*, *CHS3*, and *CHS4* were positively correlated to four, eight, and nine TFs, respectively, and have no negative

relationships. However, *CHS2* was positively correlated to 13 TFs and negatively correlated to 10 TFs. *F3H1* showed 11 positive relationships and 8 negative relationships with TFs. *F3H3* has positive and negative relationships with eight and five TFs, respectively. *F3H4* had positive and negative relationships with 27 and 13 TFs, respectively. *F3H2* had the same regulation with *F3H4*; another two TFs (*bHLH09* and *MYB11*) also demonstrated positive relationships with *F3H2*. *DFR2* was positively correlated with 13 TFs and negatively correlated with 3 TFs (Figure 7). Overall, these results indicated that F3H2, F3H4, and UVB8-2 may be regulated by more TFs and involved in flavonoid biosynthesis and UV-B response. The TF families, namely WRKY, bZIP, MYB, bHLH, and WD40, might regulate the biosynthesis of flavonoids or their precursors according to the key functional DEGs and UVB8 demonstrating a correlational relationship with MYB, bHLH, and WD40, suggesting that the function of ternary protein complexes (MBW, MYB–bHLH–WD) might contribute to regulate flavonoid biosynthesis in *D. cambodiana*.

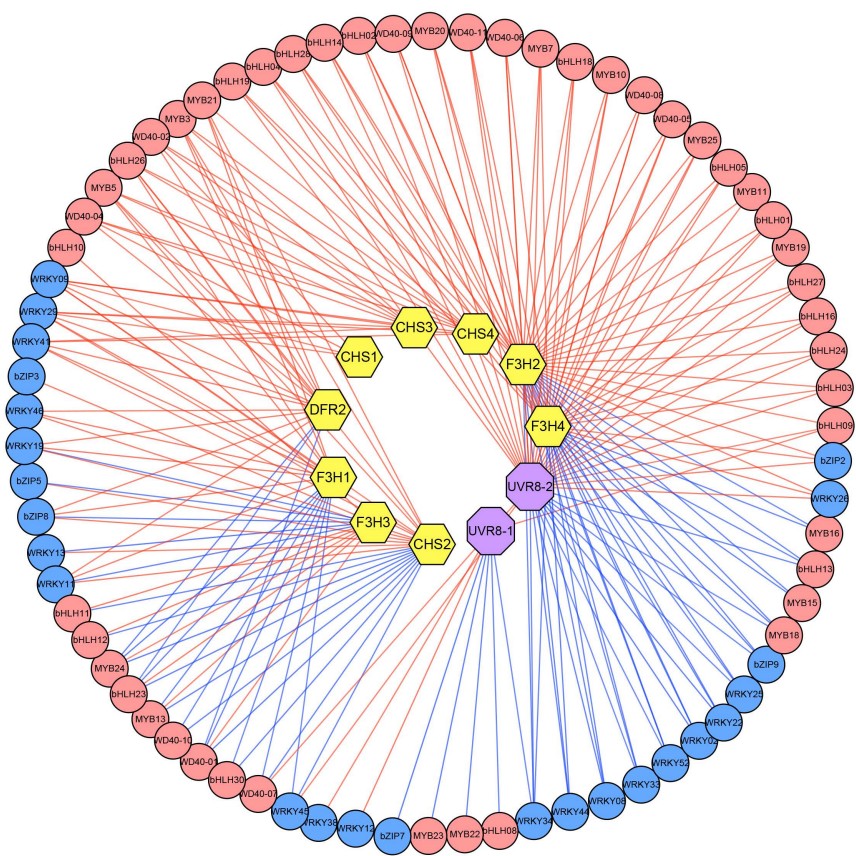

**Figure 7.** The co-expression network of DEGs in flavonoid biosynthesis, UV-B response, and transcription factors. The red lines represent positive correlations and blue lines represent negative correlations. The TFs of MBW (ternary protein complexes, MYB–bHLH–WD40) are represented by red circles. *WRKY* and *bZIP* TFs are repre nted by blue circles. The genes of UV-B response are represented by purple octagons. The genes involved in flavonoid biosynthesis are represented by yellow hexagons.

### 3.9. RT-qPCR Verification of Differential Gene Expression

To inspect the DEGs' expression pattern during UV-B irradiation, we relied on the reliability and accuracy of FPKM from transcriptomic data; a total of 12 DEGs were selected for RT-qPCR verification. Most of the selected DEGs were differentially expressed in stems of *D. cambodiana* under UV-B treatment, showing a similar profile as reflected by FPKM values (Figure 8). Therefore, this result provided accurate and reliable transcriptional pattern data for further studies on the cross-talk between UV-B response and the molecular mechanism of flavonoid formation in *D. cambodiana*.

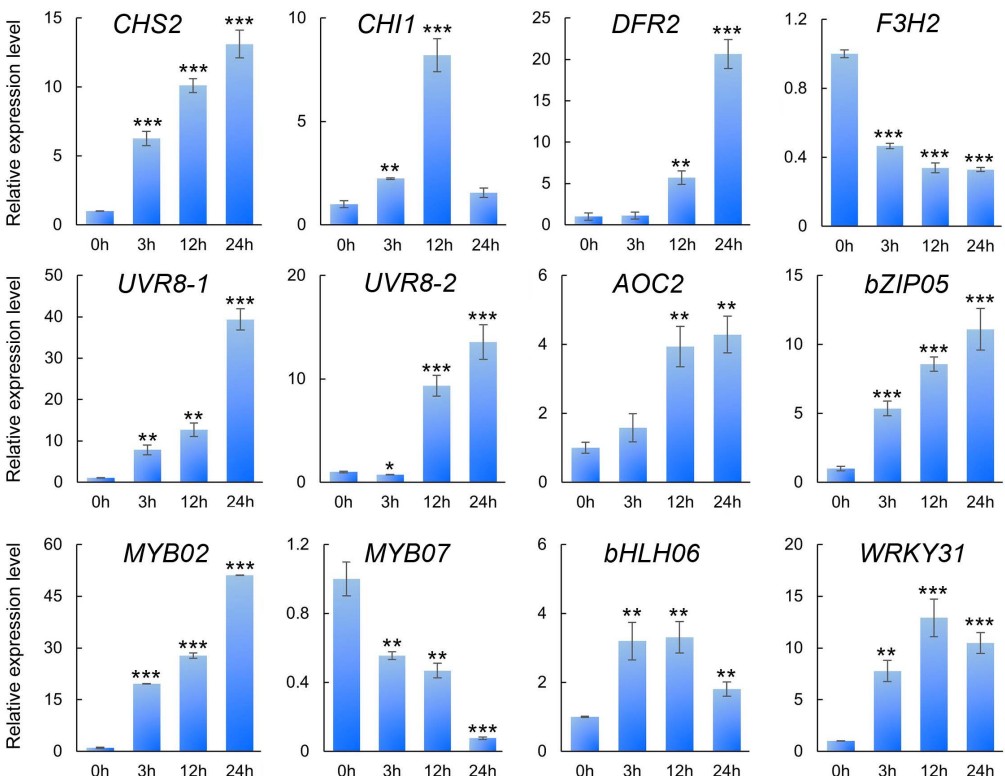

**Figure 8.** qPCR verification of 12 DEGs involved in UV-B promotion of flavonoid accumulation in *D. cambodiana*. qPCR results represent the mean (±SD) of three biological replicates. Asterisks indicate statistically significant differences when compared with 0 h samples, * $p < 0.05$, ** $p < 0.01$, *** $p < 0.001$.

## 4. Discussion

### 4.1. Effects and Damage of UV-B Radiation on Plants

UV-B has effects not only on plants, but also on their associated microorganisms. UV-B is a kind of energetic radiation often resulting in the death of microbes and a reduction in plant pathogens and phylloplane microorganism populations and persistence [52,53]. In addition, UV-B is also an important environmental signal that affects plant development and acclimation, and can be perceived by specific photoreceptors of plants to stimulate the downstream signaling cascades to activate defensive reactions [54]. ROS, as essential second messengers, can be induced by UV-B exposure, and play a pivotal role in interactions with the plant hormone pathways of salicylic acid, jasmonic acid, and ethylene [55], and lead to gene expression fluctuation and secondary metabolite production responsible for defense responses against various environment stresses [56]. However, large doses or long periods of UV-B exposure can also harm plants in the form of epidermal damage and overdose of ROS accumulation [57,58]. Regarding oxidative stress, various ROS as damaging agents can indiscriminately attack almost any cell by destructive protein modifications, mutagenic DNA strand breaks, and protein–DNA cross-links, and influence cell membrane lysis and plant plastid-structure destruction [59,60]. ROS generation from photosensitizing reactions can lead to responses to oxidative stress, which manifest as programmed cell death and apoptosis [61]. Due to the presence of phenolic hydroxyl groups in the structure of flavonoid, flavonoids usually act as scavengers of ROS and protect plant cells from membrane leaking and lysis caused by high ROS levels [59].

### 4.2. Identification of UV-B Signaling Perception and Transduction Genes in D. cambodiana

Green plants require photosynthesis to live, but they are unavoidably exposed to solar UV radiation. However, high-intensity and long-term UV-B radiation can impair multiple biological effects of plants [62]. UV RESISTANCE LOCUS 8 (UVR8), as the specific



photoreceptor of UV-B, was first identified from a mutant of *Arabidopsis* hypersensitive to UV-B. Subsequently, *UVR8* genes were characterized in green algae, bryophytes, lycophytes, and angiosperms, and chlorophytes were regarded as premier organisms from the Viridiplantae group, where UVR8 presents itself as part of plant evolution [25]. The structure and function of UVR8 have been described in various plants, such as liverwort [63], moss [64], maize [65], *Arabidopsis* [66], apple [67], and so on. The inactive UVR8 generally appears as a dimer in the nucleus and in cytoplasm [68]. Upon UV-B irradiation, the UVR8 dimer switches to a monomer and accumulates in the nucleus [69], initiating regulatory functions via associated signaling networks, gene expression, and metabolic pathways in plants [70]. UVR8 is inactivated while the monomers become redimerized with the assistance of RUP1 and RUP2 [71,72]. This is a strongly conserved mechanism in plant evolution, and UVR8 preforms an essential role in UV-B acclimatization and in the survival of plants [25]. In our study, two unigenes were identified as *UVR8*, and both showed significantly different expressions in the stems of *D. cambodiana* after UV treatment. Our preliminary conclusion was that *DcUVR8* genes help dragon trees tolerate UV-B stress with similar mechanisms. Additionally, the UV-B receptor UVR8 plays a crucial role in regulating flavonoid biosynthesis or accumulation, resulting in an enhanced tolerance to UV-B radiation in plants [73]. CmUVR8 regulates the expression of genes involved in flavonoid biosynthesis and the amassing of UV-B-induced flavonoids in *Chrysanthemum morifolium* [74]. UVR8 regulates the biosynthesis of protective flavonoids to coordinate UV-B responses in liverwort [63], and UVR8-mediated flavonoid induction to increase UV-B tolerance is regarded as a characteristic conserved across liverwort and flowering plants. Previous studies, usually on effects in plants, including the model plant *Arabidopsis*, have certified that UVR8 is up-regulated, influencing the biosynthesis of UV-B protectants, especially of flavonoids, and even regulates the transduction of UV-B perception signals downstream [71,75]. However, in our experiments, both *DcUVR8s* were down-regulated in the stems of dragon trees after UV treatment, as reported in *Camellia sinensis* and maize. *UVR8* from *C. sinensis* was inhibited after both short and long UV-B exposure periods; the down-regulation of maize *UVR8* was measured after long-term UV-B exposure [76,77]. This is in contrast with numerous studies, and it will be significative to uncover the underlying molecular mechanism of UVR8 down-regulation in dragon trees under UV-B exposure, with significantly implications regarding the accumulation of flavonoids.

### 4.3. Effect of UV-B Radiation on Flavonoid Biosynthesis

Flavonoids usually protect plants against UV-B stress and high light exposure, and it has been widely demonstrated that UV-B supplementation increases flavonoid content in various plants. The total accumulation of flavonoids and phenolics was enhanced in in vitro propagated and acclimatized plantlets of *Artemisia annua* supplied with UV-B radiation [78]. Taxoids (especially paclitaxel) and flavonoids are mainly bioactive contents of *Taxus* species, and their accumulation was significantly increased after 12 and 24 h of UV-B radiation in *T. cuspidata* plantlets [79]. The contents of flavonoids and soluble sugars were promoted, but the biosynthesis of anticancer compound podophyllotoxin was inhibited in *Sinopodophyllum hexandrum* treated with UV-B stress [80]. The exposure of lettuce (*Lactuca sativa*) to UV-B can induce quercetin flavonoid accumulation to restrict the establishment of pathogens, thus reducing the plant's susceptibility to disease [81]. The expression of structural genes involved in flavonoid synthesis were up-regulated and the biosynthesis and metabolism of flavonoids, the pharmaceutical active elements of *Ginkgo biloba* extracts, were promoted when *G. biloba* leaves suffered long-term UV-B exposure [82]. Double the content of baicalin was measured in *Scutellaria baicalensis* roots treated with suitable concentrations of UV-B for 24 h during the postharvest drying process [83]. In the present work, the stems of *D. cambodiana* were treated with UV-B radiation, resulting in significantly enhanced total flavonoid accumulation (Figure 1d).

All flavonoids result from the phenylpropanoid pathway, which is also in charge of the generation of many other significant secondary metabolites in plants, including lignin,

chlorogenic acids, and phenolic glycosides. Phenylalanine is converted into 4-coumaroyl-CoA catalyzed by PAL, C4H, and 4CL, then participates in the flavonoid biosynthesis pathway [84]. *CHS*, *CHI*, *F3H*, and *FLS* are responsible for the generation of common precursors and are termed early biosynthetic genes (EBGs). Correlatively, the later downstream genes (i.e., *DFR* and *LAR*) are termed late biosynthetic genes (LBGs), and are responsible for different chemical modifications to the basic flavonoid skeleton [33,85]. Previous studies have speculated that flavonoids are synthesized through the phenylpropanoid pathway, transforming phenylalanine into 4-coumaroyl-CoA, which finally enters the flavonoid biosynthesis pathway [13]. Flavonoid biosynthesis is a complicated and volatile process involving a series of enzymes, including phenylalanine ammonia lyase (PAL), cinnamate-4-hydroxylase (C4H), 4-coumaroyl CoA ligase (4CL), chalcone synthase (CHS), chalcone isomerase (CHI), flavonoid synthase (FLS), and flavonoid 3-hydroxylase (F3'H) [39]. PAL, C4H, and 4CL enzymes are important regulators in the synthesis of the phenylpropane metabolic pathway [86]. In this research, four *PAL*, one *C4H,* and fourteen *4CL* DEGs were identified from the transcriptome data. The expressions of *PAL*, *C4L,* and *4CH* DEGs in the stems of *D. cambodiana* were up-regulated significantly after UV treatment. After UV treatment, the key enzyme expression of the phenylpropane metabolic pathway was up-regulated significantly, indicating that the upstream phenylpropane metabolic pathway was enhanced. In addition, all of the DEGs encoding key enzymes of flavonoid biosynthesis showed significantly up-regulated expression after UV treatment, namely *CHS* (fiyr DEGs), *CHI* (one DEG), *DFR* (two DEGs), *LAR* (two DEGs), *F3H* (one DEG), and *FLS* (one DEG) (Figure 5b), which was consistent with the accumulation of total flavonoids in the stems of *D. cambodiana* after UV treatment. These results indicated that the key enzymes of the flavonoid synthesis pathway were significantly expressed in the stems of *D. cambodiana* after ultraviolet-B irradiation, resulting in total flavonoid content accumulation in the same tissue.

*4.4. Transcription Factors Involved in Secondary Metabolism Synthesis*

The regulation of transcription factors as repressors and activators is essential for the fine-tuning of flavonoid biosynthesis during plant growth or in plants under (a)biotic stress. Previous studies have indicated that flavonoid biosynthesis could be regulated directly by various transcription factors, such as *R2R3-MYB*, *bHLH*, *bZIP*, and *WRKY*. *MYB134* and *MYB115* are key activators, while MYB165 is a crucial repressor of flavonoid accumulation in poplar [87,88]. It was verified that GlMYB4 and GlMYB88 could positively regulate the production of flavonoids in licorice cells [89]. Overexpression of *NtMYB3* decreased the expression levels of genes involved in anthocyanin and flavonoid biosynthesis, and resulted in the accumulation of proanthocyanin by repressing the biosynthesis of flavonoids in narcissus [90]. Three MYB (MYB11, MYB12, and MYB111) proteins, all positive regulators, could activate the expression of key genes in the flavonoid synthesis pathway and positively regulate flavonoid biosynthesis [91–93]. There are 26 *MYB* TF genes from *D. cambodiana* showing different expression profiles in response to UV-B exposure, suggesting that MYB might contribute to the flavonoid production. The combination of MYB and bHLH was shown to regulate the flavonoid biosynthesis pathway and govern flavonoid production in pomegranate [94], *Arabidopsis* [95], and *Nicotiana benthamiana* [96]. Furthermore, some MYB and bHLH proteins, together with WD40, can constitute ternary protein complexes named MBW (MYB–bHLH–WD), and play a regulatory role in flavonoid biosynthesis by specifically activating the expression of flavonoid late biosynthetic genes [95]. Furthermore, *DcbHLH5* was co-expressed with flavonoid biosynthetic genes in the UV-B exposure treatment, and DcbHLH5 participated in flavonoid biosynthesis by activating or suppressing the expression of flavonoid biosynthesis genes via interactions with their promoters [97]. In this study, we found that 30 bHLH genes were differentially expressed in the stems of *D. cambodiana* after UV treatment; they are potential regulators involved in the biosynthesis of flavonoids in *D. cambodiana.* Further work will focus on determining whether the combination of MYB and bHLH or ternary MBW TFs are responsible for the flavonoid accumulation of dragon's blood or not. The functional relevance of *Arabidopsis'* elongated

hypocotyl 5 (HY5, bZIP TFs from Group H) in flavonoid production and UV-B photoreceptor processes is well-established [98]. HY5 mediates UV-B-induced changes in gene expression downstream of the photoreceptor UVR8 and contributes to the establishment of UV-B tolerance in *Arabidopsis* [99]. HY5 is necessary to the transcriptional activation of the multiple flavonoid biosynthesis genes in *Arabidopsis* [100,101]. Overexpression of *NtHY5* positively regulates the expression of phenylpropanoid pathway genes with the benefit of an increase in flavonoid content in tobacco, and also contributes to the establishment of UV-B tolerance [102]. UV-B exposure led to the down-regulation of two *bZIPs*, and to the up-regulation of seven *bZIPs* in *D. cambodiana* (Figure 6), indicating that *bZIPs* may be those DEGs in the dragon tree responding to the UV-B challenge and regulating the production of flavonoids in dragon's blood. *WRKY* genes are also regulators of flavonoid accumulation in cotton [103], *Freesia hybrida* [104], *Camellia sinensis* [105], and so on. Some genes from flavonoid biosynthesis were regulated by *OsWRKY89* in rice [106]. *MdWRKY72* increased anthocyanin synthesis by directly and indirectly promoting *MdMYB1* expression in apple plants [107]. A total of 63 *WRKY* DEGs were found as different expressions in the stems of *D. cambodiana* after UV treatment, more than other TF families. These results imply that WRKY DEGs may also perform significant regulatory roles in flavonoid biosynthesis and resistance to UV-B radiation in *D. cambodiana*. Although MYB, bZIP, bHLH, and WRKY TFs involved in regulating flavonoid accumulation have been well-characterized in various plants, the related TFs are poorly studied in *D. cambodiana*. More studies are still needed to further characterize transcription factors, flavonoid biosynthesis, and UV-B radiation in *D. cambodiana*, and to determine whether combinations of TFs complexes are responsible for the flavonoid accumulation of dragon's blood or not.

### 4.5. The Potential Mechanism of UV-B Promotes Flavonoid Accumulation in D. cambodiana

　　Nitric oxide (NO) is a crucial signaling molecule involved in stomatal regulation and plant response to abiotic stress and pathogen infection [108]. Plant response to UV stress was regulated by NO [109]. Previous studies have demonstrated that plant stress enhances NOS activity and NO content, and mediates the UV-B radiation-induced expression of specific genes (such as chalcone synthase CHS) [110], in which $Ca^{2+}$ concentration increases and MAPK regulates NO production [111,112]. In addition, the main signaling molecules triggered by ultraviolet radiation are ROS, and they, together with ethylene and jasmonic acid (JA), occasion the expression of phenylpropanoid genes, resulting in the activation of the PAL enzyme (CHS and PAL are key enzymes in flavonoid biosynthesis [113]). Plants under (a)biotic stress produce reactive oxygen species (ROS); hydrogen peroxide ($H_2O_2$) and nitric oxide (NO) are typical ROS. Recently, studies indicated that ROS are also a signaling transduction factor in plants, but ROS might injure the plant itself, especially the plant cell membrane [59].

　　Finally, the potential mechanism of UV-B-promoting flavonoid accumulation in *D. cambodiana* was deduced by combining previous research with this study (Figure 9). UV-B can damage the dragon tree via reactive oxygen species (ROS), and one typical reactive oxygen, hydrogen peroxide ($H_2O_2$), is a signal factor in plant response and activates three approaches to protect the dragon tree. The prime jasmonic acid pathway and other plant hormone pathways activating plant defense responses and another two modules may promote flavonoid synthesis. In these two promotional flavonoid pathways, hydrogen peroxide could directly induce transcriptional factor expression via kinase cascades and could also indirectly induce TF expression via calcium ion levels, but this approach should be bridged with NO signaling. Following the TF-regulated expression of genes involved in flavonoid biosynthesis, the flavonoids and their derivatives may be synthesized; flavonoids scavenge the reactive oxygen as antioxidants to protect the dragon tree, thus avoiding ROS damage.

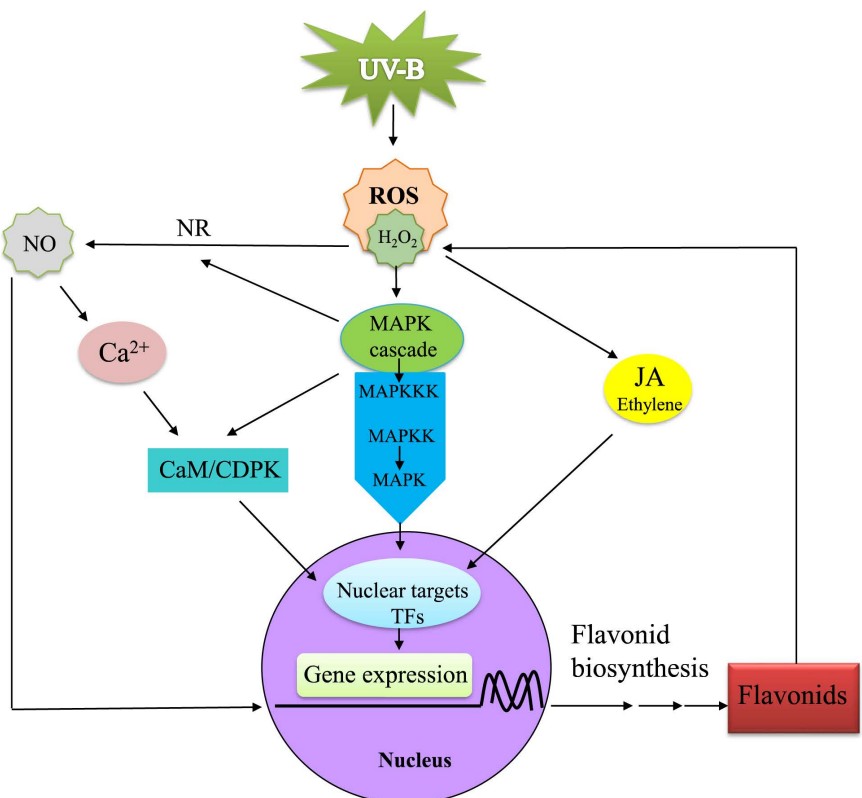

**Figure 9.** Potential mechanism of UV-B promoting flavonoid accumulation in *D. cambodiana*.

## 5. Conclusions

The total flavonoid content increased significantly in the stems of *D. cambodiana* after UV-B exposure. RNA-seq analysis provides new insight into UV-B radiation promoting flavonoid accumulation in *D. cambodiana*. The functional enrichment analysis indicated that some DEGs were related to secondary metabolisms, such as those of phenylpropanoids, flavonoids, stilbenoids, diarylheptanoids, gingerol, sesquiterpenoids, and triterpenoids. A total of 256 core DEGs were further analyzed and included genes related to UV response, flavonoid biosynthesis, and transcription factors. The co-expression network suggested that the TFs might mediate the process from UV-response and flavonoid biosynthesis, especially concerning the ternary protein complex, MBW. Finally, the potential mechanism of UV-B-promoting flavonoid accumulation in *D. cambodiana* was deduced based on transcriptomic analysis, a co-expression network, and previous studies from other plants. These results enhanced the understanding of the dragon tree's adaptation to UV radiation and provided genetic resources for the conservational biology of the dragon tree.

**Supplementary Materials:** The following supporting information can be downloaded at https://www.mdpi.com/article/10.3390/f14050979/s1, Table S1: Primers for quantitative analysis of gene expression; Table S2: Summary of transcriptome data of *D. cambodiana* after UV-B exposure; Table S3: Transcriptome data—assembled transcripts and Unigene statistics; Table S4: GO enrichment analysis; Table S5: KEGG enrichment at 3 h vs. 0 h; Table S6: KEGG enrichment at 12 h vs. 0 h; Table S7: KEGG enrichment at 24 h vs. 0 h; Table S8: DEGs involved in plant UV response; Table S9: FPKM values of DEGs involved in signal cascade transduction; Table S10: FPKM value of DEGs in flavonoid biosynthesis; Table S11: FPKM of TF-,TR-, and PK-related unigenes.

**Author Contributions:** Conceptualization, X.D. and H.D.; methodology, X.D.; software, H.Z. and X.D.; validation, Y.-E.L. and H.Z.; formal analysis, Y.-E.L. and J.Z.; investigation, H.W.; resources, J.Z.; data curation, B.J.; writing—original draft preparation, Y.-E.L., H.Z. and X.D.; writing—review and editing, X.D. and W.M.; visualization, Y.-E.L. and H.Z. and Y.-E.L.; supervision, X.D. and H.D.; project administration, X.D.; funding acquisition, X.D. and H.D. All authors have read and agreed to the published version of the manuscript.

**Funding:** This research was supported by the National Natural Science Foundation of China (81803663), the Central Public-interest Scientific Institution Basal Research Fund for Chinese Academy of Tropical Agricultural Sciences (1630052020003), and the earmarked fund for China Agriculture Research System (CARS-21).

**Data Availability Statement:** The raw data of the transcriptomic analysis have been deposited at NCBI with BioProject PRJNA427675, and unigene sequences have been deposited at Figshare with the doi:10.6084/m9.figshare.22180546.

**Acknowledgments:** We would like to acknowledge the invitation from the Second Dragon Tree Conference and Petr Maděra who suggested our topic in the content of the Special Issue "New Knowledge in Dragon Tree Research" in *Forests*.

**Conflicts of Interest:** The authors declare no conflict of interest.

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
