# Peer review of "Transcriptomic Analysis Reveals the Involvement of Flavonoids Synthesis Genes and Transcription Factors in Dracaena cambodiana Response to Ultraviolet-B Radiation"

_forests, doi:10.3390/f14050979_

Round 1

Reviewer 1 Report

1. Throughout the article, certain sections contain complex descriptions and technical terms that might make it difficult for readers to comprehend. For example, in the section discussing transcription factors, numerous gene names and their roles are mentioned, which can be overwhelming.

2. The article does not provide detailed information on the replication of experiments between different treatment groups, which may impact the reliability of the results. For instance, when discussing the total flavonoid content increase after UV-B exposure, there is no mention of replicates or the number of samples analyzed.

3. Some evidence in the article is based on references from other studies, without independent validation. For example, the roles of MYB, bHLH, bZIP, and WRKY transcription factors in regulating flavonoid accumulation are largely based on previous research in other plant species, with limited experimental validation in D. cambodiana.

4. The article mentions differential gene expression for certain genes but does not explicitly state the specific impact of these differences on the physiological response to UV-B radiation.

5. Some conclusions in the article are based solely on transcriptomic data and co-expression network analysis, which may introduce biases. For example, the proposed mechanism of UV-B promoting flavonoid accumulation in D. cambodiana is primarily based on these analyses, without direct experimental validation.

6. As you pointed out, the article does not provide direct evidence of the physiological response in the plant to the gene expression changes under UV-B radiation. This makes it difficult to confirm whether the gene expression changes indeed lead to the expected physiological response. Additionally, the lack of analysis of H2O2 further complicates the confirmation of the proposed mechanism in Figure 9.

To address these concerns, the authors could consider simplifying complex descriptions for better readability, providing more details on experimental replicates, performing independent validation of key evidence, clarifying the specific impact of differentially expressed genes, and conducting additional experiments to directly examine the physiological response of the plants to UV-B radiation and the role of H2O2 in the proposed mechanism.

Author Response

Dear Reviewer,

Thanks for your comments concerning our manuscript entitled “Transcriptomic Analysis Reveals the Involvement of Flavonoids Synthesis Genes and Transcription Factors in Dracaena cambodiana Response to Ultraviolet-B Radiation” (ID: Forests-2279599). Those comments are all valuable and very helpful for revising and improving our paper, as well as the important guiding the significance to our future research. We have studied the comments carefully and have made correction which we hope meeting with approval. Revised positions are marked in red in the manuscript with track changes. The main corrections in this revised manuscript and the responds to the reviewer comments are as the following:

Comment 1. Throughout the article, certain sections contain complex descriptions and technical terms that might make it difficult for readers to comprehend. For example, in the section discussing transcription factors, numerous gene names and their roles are mentioned, which can be overwhelming.

Response: This manuscript is a typical analysis about plant transcriptome. The mechanism of plant flavonoids synthesis and regulation are complex and this complexity was enhanced by the UV-B intervene. Actually, the extremely large amounts of genes were obtained from the transcripts assembly and we just focused on the genes involved in flavonoids synthesis and transcription factors (TF) related to the flavonoids synthesis (especially for the ternary complex of MYB-bHLH-WD40, bZIP and WRKY) in D.cambodiana treatment with UV-B, the other genes or TFs were not analyzed in this manuscript. The names and roles of genes and TFs in the manuscript have been reduced before first uploaded and it might be impacted the integrity and systematic of manuscript if the description were furtherly retrenched or optimized.

Comment 2. The article does not provide detailed information on the replication of experiments between different treatment groups, which may impact the reliability of the results. For instance, when discussing the total flavonoid content increase after UV-B exposure, there is no mention of replicates or the number of samples analyzed.

Response: Thanks for your suggestion and all replicates of treatments have been added in the revised manuscript at the Line 166 to Line 168 and Line 456 to Line 457.

Comment 3. Some evidence in the article is based on references from other studies, without independent validation. For example, the roles of MYB, bHLH, bZIP, and WRKY transcription factors in regulating flavonoid accumulation are largely based on previous research in other plant species, with limited experimental validation in D. cambodiana.

Response: Transcription factors (TFs) play the crucial roles in up or down regulating the functional genes expression via promoter sequences of functional genes. The nucleotide sequences of promoters were usually gained from the whole genome sequences of species and the D.cambodiana genome is lacked until now, so the regulation mechanism have not been directly performed at this step. But D.cambodiana is still the excellent species for the scientific research about UV-B tolerance of plant according to their survival environments, the new abiotic resistance genes might be mined from this specie once its genome were obtained and this manuscript is preliminarily identified some key TFs and their expression profiles in D.cambodiana based on the other plant research for guilding our current research about ecological significance of dragon trees and provide basic data for the future research about dragon trees.

Comment 4. The article mentions differential gene expression for certain genes but does not explicitly state the specific impact of these differences on the physiological response to UV-B radiation.

Response: Thanks for your suggestion. In the revised manuscript, the content of H2O2 and the activities of CAT and POD involved in H2O2 or ROS scavenge were tested. Revised text in the Line 232 to Line 243.

Comment 5. Some conclusions in the article are based solely on transcriptomic data and co-expression network analysis, which may introduce biases. For example, the proposed mechanism of UV-B promoting flavonoid accumulation in D. cambodiana is primarily based on these analyses, without direct experimental validation.

Response: This comment is pretty suggestion for research about rare species. Transcriptome and its derived co-expression network construction are the common methods for promote the research of rare species into the field of molecular biology, these methods will provide useful resources of genes and regulation network for predicting potential TFs interactive with the functional genes by lacking whole genome data. In addition, the previous studies from model plant have been suggested that UV-B radiation could promote ROS burst and H2O2 might be the signal factor. The dragon blood, as the special resin from dragon trees, contained abundant flavonoids and its derivatives, and these compounds showed excellent antioxidant activities based the previous studies from our group or other researchers. Combining with the previous studies from other plant, especially about model plant, the relatively accurate regulatory mechanism will be proposed, but this regulatory mechanism is not static and will be perfected via further studies from the new data published and more researchers participate in study of dragon tree.

Comment 6. As you pointed out, the article does not provide direct evidence of the physiological response in the plant to the gene expression changes under UV-B radiation. This makes it difficult to confirm whether the gene expression changes indeed lead to the expected physiological response. Additionally, the lack of analysis of H2O2 further complicates the confirmation of the proposed mechanism in Figure 9.

Response: Thanks for your suggestion. As the response to Comment 4, the content of H2O2 and the activities of CAT and POD involved in H2O2 or ROS scavenge were tested and added in the revised manuscript. We hope these parameters will neutralize this refuse.

Comment 7. To address these concerns, the authors could consider simplifying complex descriptions for better readability, providing more details on experimental replicates, performing independent validation of key evidence, clarifying the specific impact of differentially expressed genes, and conducting additional experiments to directly examine the physiological response of the plants to UV-B radiation and the role of H2O2 in the proposed mechanism.

Response: Thanks for your suggestion. Please refer the above responses about comment 1~6 to answer this comprehensive comment.

We tried our best to improve the manuscript and made some additional experiments or changes in the revised manuscript. These experiments and changes will not influence the content and framework of the paper. And here we did not list the changes but marked in red in revised paper. We appreciated for you warm work earnestly, and hope that the correction will meet with approval. Once again, thanks for your comments and suggestion.

Sincerely,

Haofu Dai and Xupo Ding

Key Laboratory of Research and Development of Natural Product from Li Folk Medicine of Hainan Province, Institute of Tropical Bioscience and Biotechnology, Chinese Academy of Tropical Agricultural Sciences, Rd. Xueyuan No.4, Haikou, 571101, China.

Tel: +86-89866968572; E-mail: daihaofu@itbb.org.cn; xupoding@hotmail.com

Reviewer 2 Report

Title. The title is appropriate to the subject, informative, and concise. This answers the three important questions: What? Where? How?

Abstract. The abstract is concise, provides a clear overview, includes essential facts for the paper, and concludes with a final point that places the work described in a broader context.

Keywords. These are enough for the topic.

Introduction. The introduction includes background to provide an appreciation for the context of the work presented and also states the rationale and problem that the researchers attempted to answer through their experiments. In this section, the authors talk about flavonoids, ultraviolet-B (UV-B), D. cambodia, and the purpose of the research.

Material and methods.  In this section, the authors describe the correct steps that were followed while conducting their study and explain how they analyzed the data. The methods were well implemented to reach the objectives proposed.

Results. This section was well written and shows all data with good descriptions. The results say about the objective that motivates the research, and the authors take a broad look at their findings and examine the work in the larger context of the field.

L248 to L249 — Change the order on the paper. First, stand Figure 2.

L274 to L307 — Figures 4a–4f must be Figures 3a–3f.

L314 to L317 — These two sentences must be moved to the discussion section.

L331 to L344 — The paragraph and the last sentence discuss some previous results. I suggest moving it to the discussion section.

Discussion. The authors had to discuss the data with respect to how their data fit into what is currently known in the field.

Conclusion. This section included the major conclusions, which were briefly written.

Figures and Tables. Both sections have good information and are necessary for the manuscript, they depict the data nicely.

Figures 1  and 8 — What do mean the asterisks (**, ***)? Explain it.

Figure 2b — Write the scientific names in italics.

Figure 3 — The letters are very small. Increase their size.

Figure 6 — It was set in the results section but wasn't cited.

Author Response

Dear Reviewer,

Thanks for your notification about our manuscript entitled “Transcriptomic Analysis Reveals the Involvement of Flavonoids Synthesis Genes and Transcription Factors in Dracaena cambodiana Response to Ultraviolet-B Radiation” (submission ID: Forests-2279599) submitted to the journal of Forests. Those comments are all valuable and very helpful for revising and improving our manuscript. We have made some changes according to your comments and suggestions. Revised positions are marked in red in the manuscript with track changes. The responses of point to point are as the followings:

Comment 1. Title. The title is appropriate to the subject, informative, and concise. This answers the three important questions: What? Where? How?

Response: Thanks for your approval about the title of this manuscript.

Comment 2. Abstract. The abstract is concise, provides a clear overview, includes essential facts for the paper, and concludes with a final point that places the work described in a broader context.

Response: Thanks for your approval about the abstract of this manuscript.

Comment 3. Keywords. These are enough for the topic.

Response: Thanks for your approval about the keywords of this manuscript.

Comment 4. Introduction. The introduction includes background to provide an appreciation for the context of the work presented and also states the rationale and problem that the researchers attempted to answer through their experiments. In this section, the authors talk about flavonoids, ultraviolet-B (UV-B), D. cambodiana, and the purpose of the research.

Response: Thanks for your approval about the introduction of this manuscript.

Comment 5. Material and methods.  In this section, the authors describe the correct steps that were followed while conducting their study and explain how they analyzed the data. The methods were well implemented to reach the objectives proposed.

Response: Thanks for your approval about the material and methods of this manuscript.

Comment 6. Results. This section was well written and shows all data with good descriptions. The results say about the objective that motivates the research, and the authors take a broad look at their findings and examine the work in the larger context of the field.

Response: Thanks for your approval about the results of this manuscript and the response about the details of comments are as the following:

L248 to L249 — Change the order on the paper. First, stand Figure 2.

Response: This suggestion has been revised at Line 276 to Line 283 in the revision.

L274 to L307 — Figures 4a–4f must be Figures 3a–3f.

Response: This mistakes were revised at Line 306 to Line 325 in the revision.

L314 to L317 — These two sentences must be moved to the discussion section.

Response: These sentences were repetitive with the paragraph at the Line 483 to Line 492, so we delete them in the revision. 

L331 to L344 — The paragraph and the last sentence discuss some previous results. I suggest moving it to the discussion section.

Response: These sections have been moved to the discussion at Line 614 to Line 627 in the revision.

Comment 7. Discussion. The authors had to discuss the data with respect to how their data fit into what is currently known in the field.

Response: Thanks for your suggestion. This manuscript might be a first report about dragon tree response to the UV-B stress. In each paragraph of discussion, we firstly present progress of the previous or current research in each section, then the compared to our results and discussed at the end of each section in the manuscript. The molecular biological research about dragon trees were limited until now and the dragon tree interactived with UV-B was not present in the previous studies. So we have been compared our results with the similar research from other plant.

Comment 8. Conclusion. This section included the major conclusions, which were briefly written.

Response: Thanks for your approval about the conclusion of this manuscript.

Comment 9. Figures and Tables. Both sections have good information and are necessary for the manuscript, they depict the data nicely.

Response: Thanks for your approval about the figures and tables of this manuscript and the response about the details of comments are as the following:

Figures 1 and 8 — What do mean the asterisks (**, ***)? Explain it.

Response: The asterisks represent the significant statistics with the ANOVA test between treatments and control groups at 0 hour. Two asterisks represent significant difference with P<0.01, three asterisks represent significant difference with P<0.001 and four asterisks represent significant difference with P<0.0001. These notes have been added in the revision at Line 224 to Line 225 and Line 457.

Figure 2b — Write the scientific names in italics.

Response: The Latin names have revised as italics.

Figure 3 — The letters are very small. Increase their size.

Response: The GO enrichment of DEGs contained more information and the letters size increased will result the letters overlap, so we increased scale proportion about the GO enrichments (Figure 3a, 3c and 3e) in the Figure 3 in the revision.

Figure 6 — It was set in the results section but wasn't cited.

Response: Figure 6 was cited in the revision at Line 407 and Line 599.

We tried our best to improve the manuscript and made some changes in the revised manuscript. These changes will not influence the content and framework of the paper. And here we did not list the changes but marked in red in revised manuscript. We appreciated for you warm work earnestly, and hope that the correction will meet with approval. Once again, thanks for your comments and suggestion.

Sincerely,

Haofu Dai and Xupo Ding

Key Laboratory of Research and Development of Natural Product from Li Folk Medicine of Hainan Province, Institute of Tropical Bioscience and Biotechnology, Chinese Academy of Tropical Agricultural Sciences, Rd. Xueyuan No.4, Haikou, 571101, China.

Tel: +86-89866968572; E-mail: daihaofu@itbb.org.cn; xupoding@hotmail.com

Round 2

Reviewer 1 Report

  1. The author provided a complete experimental method, greatly improving the study's methodological transparency.
  2. However, there was not enough discussion or response to the issues raised in the previous review.
  3. Line 613: Section 4.5, "The proposed mechanism of UV-B promotes flavonoids accumulation in D. cambodiana," should be the main point of this study. However, as previously mentioned, this section did not apply to this study's results or provide enough evidence to support the conclusion presented in Figure 9. Without relevant evidence to support this conclusion, it is difficult to evaluate the manuscript's importance and research value. The author needs to provide complete and direct evidence to explain the relationship between this conclusion and the previous research. Otherwise, this study is merely reconfirming previous research, without proposing new perspectives.

Author Response

Dear Reviewer,

Thanks for your additional comments and approval about the revised methods in our revised manuscript entitled “Transcriptomic Analysis Reveals the Involvement of Flavonoids Synthesis Genes and Transcription Factors in Dracaena cambodiana Response to Ultraviolet-B Radiation” (ID: Forests-2279599). Those comments are valuable for plant research and very helpful for revising and improving our manuscript, as well as the important guiding the significance to all future research about non-model plants, especially for the rare plants. We have studied the comments carefully with the senior scholars in our institute and have made correction which we hope meeting with approval. New revised positions are marked in blue in the manuscript with track changes. The main corrections in this revised manuscript and the responds to the reviewer comments are as the following:

Comment 1. The author provided a complete experimental method, greatly improving the study's methodological transparency.

Response: Thanks for your approval about the methods in the revised manuscript.

Comment 2. However, there was not enough discussion or response to the issues raised in the previous review.

Response: We studied the previous comments again and speculated that physiological response from Comment 4 and Comment 6 were not enough respond. In this study, we focused on the plant stem prior response to UV-B radiation and this was the plant emergency reaction. For the early response in plants, variation of nearly all genes expression were significantly presented by transcriptome and this was a directly genetic response, which will guide the following physiological response. The activity of main enzyme was the representative physiological response. According to the Comment 4, we firstly detected the contents of H2O2 and subsequently tested the activities of catalase (CAT) and peroxidase (POD), which were the main enzymes in scavenging H2O2 in plant cell. CAT could directly catalyze the H2O2 into H2O and oxygen, but the CAT activity was not significantly different in our treatments suggesting that POD might play the important role in scavenging H2O2. POD could not directly decompose H2O2 but indirectly complex reaction between H2O2 and flavonoids into complexes and oxygen, and these complexes will feedback inhibited the activity of POD. We will improve this manuscript if the particular and concrete methods of physiological response index were suggested by reviewers and editors.

Comment 3. Line 613: Section 4.5, "The proposed mechanism of UV-B promotes flavonoids accumulation in D. cambodiana," should be the main point of this study. However, as previously mentioned, this section did not apply to this study's results or provide enough evidence to support the conclusion presented in Figure 9. Without relevant evidence to support this conclusion, it is difficult to evaluate the manuscript's importance and research value. The author needs to provide complete and direct evidence to explain the relationship between this conclusion and the previous research. Otherwise, this study is merely reconfirming previous research, without proposing new perspectives.

Response: This manuscript was aimed to mine the DEGs involved in UV-B stress response, flavonoids biosynthesis and regulation in dragon tree exposed in UV-B radiation, gain some insight into the relationship between Dracaena species and UV-B radiation. The section 4.5 and Figure 9 were the potential inferences based on our study and previous research in other plants, it is not the main point of this study whereas that might be our prospective and future research according to that the mechanism could not be completely elucidated with a phase study, so this potential mechanism might be optimized by more and more researcher participating into this research. This confusion and suggestion of Comment 3 might be the common predicament in rare plant research. The interactive mechanism of plant and UV-B were still not illuminated thoroughly in plants, even in Arabidopsis thaliana. In this section 4.5 and Figure 9, genes involved in each step could be found in Figure 4~6 and expression profiles of some representative genes were also verified by qPCR in Figure 8. We have been consulted with some senior scholars in our institute and they also suggested that the extra complete and direct evidences could not got under current scientific conditions by lacking whole genome of D.cambodiana and genetic transformation system. But we keep the opening suggestion yet to improve this manuscript if the particular and concrete methods were suggested by reviewers and editors.

  About the importance and value of this manuscript were ascribable to provide the theoretical foundation for the further study of pathways of flavonoid biosynthesis in D. cambodiana, the roles of transcription factors in dragon’s blood formation and dragon tree response to UV-B stress, enhance our understanding the survival and longevity mechanism of Dracaena species and conservation biology of dragon tree. In addition, the transcriptome research about the rare plant will provide new functional and resistance gene resources for the crop improvement. The other value was that we provide the transcriptomic expression profiles and enzyme activities in the stem of D.cambodiana under the UV-B treatment whereas that most of other research were focused on plant leaf exposed in UV-B.

We have been tried our best to improve the manuscript and made esential changes in the new revised manuscript. These changes will not influence the content and framework of the manuscript. And here we did not list the new changes but marked in blue in new revised manuscript. We appreciated for you warm work earnestly, and hope that the correction will meet with approval. Once again, thanks for your comments and suggestion.

Sincerely,

Haofu Dai and Xupo Ding

Key Laboratory of Research and Development of Natural Product from Li Folk Medicine of Hainan Province, Institute of Tropical Bioscience and Biotechnology, Chinese Academy of Tropical Agricultural Sciences, Rd. Xueyuan No.4, Haikou, 571101, China.

Tel: +86-89866968572; E-mail: daihaofu@itbb.org.cn; xupoding@hotmail.com